# Exploring supportive roles for global north investigators in global health research

Jade Pearce[1], Deepa Rao[1,2]*, Rae Jean Proeschold-Bell[3], Bethany Hedt-Gauthier[4], Keshet Ronen[2], James Pfeiffer[2], Kearsley A. Stewart[3], Joao Vissoci[3], Bryan J. Weiner[2], L Gayani Tillekeratne[3,5], Jenell Stewart[6]

1 Office of the Dean, School of Public Health, University of Washington, Seattle, Washington, United States of America, 2 Department of Global Health, Schools of Public Health and Medicine, University of Washington, Seattle, Washington, United States of America, 3 Duke Global Health Institute, Duke University, Durham, North Carolina, United States of America, 4 Blavatnik Institute Global Health and Social Medicine, Harvard Medical School, Boston, Massachusetts, United States of America, 5 Division of Infectious Diseases, Duke University, Durham, North Carolina, United States of America, 6 Division of Infectious Diseases, Hennepin Healthcare, Minneapolis, Minnesota, United States of America

* deeparao@uw.edu

## Abstract

Many academics are increasingly pushing for solutions to address inequitable partnership dynamics in global health research and practice. Efforts to improve equity in collaborations have prompted academic institutions to grapple with promotion and tenure criteria (usually Global North institutions), as these criteria often require leadership on grants, publications, and conference presentations for advancement. From inequitable funding opportunities to the invisibilization of Global South researchers, these components are rooted in normalizing and upholding unequal power dynamics. Therefore, the purpose of this project was to explore ways in which Global North academic institutions can value supportive roles taken by their investigators in global health research. A special session was held at the 2022, Consortium of Universities for Global Health (CUGH) meeting, entitled "Incorporating Anti-Racism, Anti-Colonialism, and Anti-Oppression Efforts into Faculty Promotion." The purpose of the workshop was to assess current practices that participants' institutions incorporated in promoting anti-racist, anti-colonialist, and anti-oppression efforts within their faculty promotion practices as well as to brainstorm future solutions. A qualitative analysis of the discussion transcripts yielded several themes, including (1) making structural changes to the promotion process, (2) incorporating anti-racism/anti-colonialism perspectives into promotion packets, (3) recalibrating agenda setting and collaboration in Global North-South partnerships, (4) assuring strong mentorship for assembly of the promotion packet, and (5) developing effective capacity strengthening efforts. Given these results, we call upon the global health academic community to implement these suggestions in their policies and practices.

## Introduction

In 2019, a piece was published in the U.S. National Public Radio blog, *Goats and Soda*, outlining the lack of acknowledgement of Dr. Jean-Jacques Muyembe's contributions and leadership role

**Data availability statement:** The data in this study was comprised of discussions in a satellite session of a conference. The session recording is publicly available at https://cugh.confex.com/cugh/2022/meetingapp.cgi/Session/1509.

**Funding:** The authors received no specific funding for this work.

**Competing interests:** The authors have declared that no competing interests exist.

in discovering and studying the Ebola virus in the 1970s. At the time, Global North researchers were credited with the discovery, which catapulted them towards further grant funding, prominence and standing in public health and medicine, and leadership careers in global health; Muyembe, on the other hand, was "written out of history [1]." Unfortunately, stories like these are common within the global health community. Prior to 2019, many investigators in global health had begun to question academic structures that advantage Global North investigators over Global South investigators [2,3,4]. This has only been accentuated by the 2020 global reckoning with systems of racism embedded within societies and academic communities. Aspects of racism and colonialism have been frequently documented and labelled "parachute research," where a global health investigator (usually from the North) "drops into a country," conducts primary research by utilizing local resources, and returns home to submit an article for a prestigious journal [5,6,7]. Dr. Seye Abimbola, from University of Sydney, advocated for recognition of this 'Foreign Gaze' that occurs with journal authorship in global health, and called for a questioning of investigator positionality (questioning whether the 'foreign gaze' is necessary, inconsequential, corrupting) in allocating authorship order [8]. These same questions can be asked about leadership, publications and roles in Global North-South collaborative projects.

Increased pressure for decolonization has prompted academics to grapple with promotion and tenure criteria at Global North institutions, as these criteria often motivate Global North dominance in the global health arena. Generally speaking, Global North academic institutions set criteria for promotion and tenure of faculty, often through a shared governance process, such that faculty are required to hold leadership positions on publications, funding, and conference presentations for professional advancement. Promotion and tenure files are then evaluated and voted upon by peers based on these criteria, and promotion and tenure is awarded once academic administrators (e.g. deans, provosts) concur with faculty evaluations. However, these criteria are often set in Global Health fields without the inputs of global partners, ultimately at the expense of Global South investigators [9]. In addition, promotion and tenure criteria were not developed with broader structural factors in mind, such as inequalities in accessing research funds and inequalities in the types of roles local investigators play on the ground where global health research is conducted [10].

A bibliometric analysis was conducted by Adam et al. and demonstrated that Global South first authorship represents only 4% of publications listed under health policy and system research [11]. Bibliometric studies have also consistently shown that Global South researchers are listed as authors at all, let alone as first authors, despite the amount of work they contribute and given that much of the work is in their locale [12,13]. These studies have been replicated and show Global South first authorship is consistently behind those of Global North first authorship, despite some recent increase [10,11]. Further, Adam and colleagues included a survey to assess funding opportunities, and Global South respondents consistently reported stagnant funding from their local governments ranging from 11% to 15% [9]. In other words, Global South investigators must rely on international or bilateral mechanisms for much of their funding. These factors undermine equitable partnerships in global health research, creating asymmetrical resource and power dynamics [14]. Together with soft money academic structures (such as federal grants or private funding), these dynamics incentivize Global North researchers, faculty, and ultimately, universities to continue parasitic practices and to promote their own agendas while erasing the role of Global South investigators in the work [15,16].

Sometimes, collaboration efforts have reproduced unequal power relations [14]. An ideal collaboration can be defined using Samoff and Carrol's framework as:

> "a collaboration that can reasonably be expected to have mutual (though not necessarily identical) benefits, that will contribute to the development of both institutional and

*individual capacities to advance [global health] at both institutions, that respects the sovereignty and autonomy of both institutions, and that is itself empowering."[17]*

This type of collaboration is rarely present in global research partnerships, where Global North researchers have set the research agenda and Global South researchers have been recognized solely for supporting roles at best. Global South investigators are further erased through lack of recognition and the inability to attend conferences due to visa access [3,18]. Samoff and Carrol's framework in conjunction with the literature and Eichbaum et al.'s discussion of the intersection of colonialism, medicine, and global health demonstrate problematic collaboration efforts that privilege Global North investigators [19]. In essence, unequal power relations suggest colonial and racist overtones that oppress and undermine scholars from the very communities that this research affects.

These realities can be traced back to unequal power dynamics; several articles framed power dynamics through French philosopher and sociologist Pierre Bourdieu's concepts of habitus, capital, and field where "power imbalances and inequities exist at each stage of the research process [20]." From the researcher and their positionality (habitus) to the resources they bring to the country of interest (capital) and finally to the agenda setting and journal submissions (field), each step clearly demonstrates disproportionate power where Global North investigators dominate [21]. Studies have shown that Global North researchers gain significantly more opportunities professionally, including prominent authorship positions, conference presentation and attendance, and more funding support at their home institutions compared to Global South researchers [14]. Further, even the travel process (vaccines, travel fees, visa) creates costly barriers that prevent most Global South researchers from "having a seat at the table" and promoting their work at conferences and workshops [22]. Due to their positionality and opportunities presented to them, Global North researchers often set the research agenda, including what to study and how research is conducted, and receive most of the credit. Reese et al. conducted a mixed methods study to survey global health researchers from the Global North and the Global South on their perception of collaboration in pediatric research [23]. Responses from both groups suggested that:

> *"more than two-thirds of total respondents...perceived that high-income country Global North investigators had set the research agenda in previous work conducted in the Global South, with the most commonly cited reason across all respondents being that Global North investigators who had set the research agenda had access to funding."[23]*

With less funding, Global South investigators are disadvantaged in every step of the research process, and their contributions often go unrecognized. Locally driven agendas are often set by what is funded by global north funding mechanisms, which are sometimes disconnected with local priorities. An imbalance develops that supports the careers and priorities of Global North researchers at the expense of Global South researchers [24]. Therefore, the purpose of this paper is to explore ways that Global North investigators and institutions can encourage and value taking supportive, rather than leadership roles in global health research. Moving in this direction is taking a proactive stance against racism, colonialism, and oppression.

This paper outlines a range of current promotion and tenure policies academic institutions (often in the Global North) engage in and explores how these policies provide powerful incentives and barriers that can either address or reinforce oppressive practices. Further, this paper will explore ways that Global North investigators can take supportive roles in global health research by incorporating anti-racism and decolonization components into their promotion and tenure practices within global health research. We held a set of discussions with global

health professionals at an academic global health conference, exploring incentive areas of promotion and tenure, anti-racism/anti-colonialism curriculum, research agenda setting and collaboration, mentoring practices, and capacity strengthening initiatives based in the Global South.

## Materials & methods

The Consortium of Universities for Global Health (CUGH) holds an annual conference, typically in a Global North location, to address current and persistent global health challenges. For the 2022 conference, a planning committee of nine faculty from two global health institutions in the United States (US) sought to create a three-hour pre-conference session entitled, "Incorporating Anti-Racism, Anti-Colonialism, and Anti-Oppression Efforts into Faculty Promotion." The conference discussions examined current promotion and tenure policies exploring ways Global North academic institutions could change to embrace fair and equitable practices in global health through anti-racist and anti-colonial lenses. Participants' institutions paid membership fees, which then enabled session attendees to attend for free. Further, this workshop was held via an online platform, which created another incentive to easily join the discussion.

Prior to the workshop, the planning committee invited faculty with interests in diversity, equity, and inclusion from nine Global North institutions and six Global South institutions. Faculty from three Global South academic partner institutions and two Global North institutions served as co-facilitators of the session. Prior to the workshop, facilitators collected information from institutions within their academic global health networks in a survey, which sought to document current practices that incorporate anti-racism, anti-colonialism, and anti-oppression efforts into faculty promotion. The survey inquired about criteria for faculty appointment and promotion into global health academic units (e.g. teaching, grant funding, publications). Seven global health academic units responded, and facilitators compiled and presented the information provided as example promotion criteria to participants of the workshop [25]. Before submission, this study's protocol was submitted to the UW Institutional Review Board (IRB) and given a non-research determination and deemed exempt from review.

### Co-author engagement with decolonization movements

Prior to, during and after the pre-conference workshop, co-authors of this manuscript have been actively engaged in decolonization movements at their home institutions and beyond. One co-author regularly advocates for institutional change in order to create structures and policies that facilitate and incentivize Global South and Community Partner leaderships. This includes prioritizing Global South scholars in agenda setting and maximizing material benefits to Global South investigators in research projects. Another of our co-authors has a role on their local promotions committee, where they play a vital role in advising academic administrators on interpretation of promotion criteria and evaluation of their faculty. They are beginning conversations with their committees to discuss changing practices to align equity values with promotion criteria, for example, changing credit given for authorship order and grant investigator positions to honor supportive roles that are taken. After the CUGH conference, the department that several of our co-authors belonged to developed a Partner's Advisory Board (PAB), which includes leaders from the Global South and local community leaders to advise the department on how to reprioritize research, scholarship and teaching in global health. In addition, this department's faculty recently voted into current use of a new promotion and tenure handbook, that provides examples of scholarship that is community engaged and honors supportive roles with Global South partners.

## Session structure

The facilitators began the session with three initial questions: What does 'decolonizing global health' mean to you?; What do you think partnerships and academic work will look like if 'decolonized?'; What would you like [partnerships and academic work] to look like in academic global health? Using these questions as a framework, participants met first as a large group to discuss concepts related to the questions, current practices at one's university, potential best practices, and how they could be applied within the promotion and tenure process. Participants then discussed their thoughts on the group-specific questions in three small groups. Participants from Global North institutions discussed the following questions: What would you like promotion criteria to accomplish? What concerns do you have about promotion criteria currently? What would you like to change in the promotion criteria? How might we solve the problems we've identified? How can we implement accountability in written vs. unwritten rules? Participants from Global South Institutions were asked: What is your understanding of the promotion criteria in Global North universities? What concerns do you have, and how are you affected by Global North university promotion criteria? How do these solutions sound to our Global South university colleagues? Will they create new problems? Participants then returned to the large group to report out key agreed upon points from their small group, and then there was time again for full-group discussion.

The workshop was recorded, and after, the research coordinator (JP) hand transcribed the recordings from the large group sessions using a literal transcription approach. A thematic coding system was developed, in which JP noted significant words and phrases that arose from the recorded discussion. Data was considered significant if it contained at least three comments relating to a central idea. The analysis framework was aligned to Schreier and Guest et al.'s qualitative approach, and the process was repeated iteratively to capture the richness and nuance of the recorded discussion [26,27]. Part of the coding system included various code labels or short descriptions to help quickly identify key words and eventually build themes around [27]. Part of this methodology included identifying similar words and ideas within the transcription. Later, the themes were further refined by sharing with other co-authors, each of whom had attended the session, to allow for confirmation and/or differing perspectives. Survey data was collected and presented during the workshop will be included in the next sections.

# Results

## Participants

A total of 30 faculty from the following eight Global North institutions: the University of Washington, Duke University (United States), Harvard University (United States), Johns Hopkins University (United States), Stanford University (United States), Emory University (United States), University of California San Francisco (United States), and the London School of Hygiene and Tropical Medicine (UK). Additionally, three faculty were present from three Global South institutions: the University of Ruhuna (Sri Lanka), Kilimanjaro Christian Medical Center (Tanzania), and Kathmandu University (Nepal). The results presented here do not link responses with Global North or South institutions in an effort protect from identifying persons or institutions who may have provided sensitive information.

## Themes

Five broad themes emerged from the data that were closely and distally related to promotion and tenure in academic global health settings: (1) making structural changes to the promotion

process, (2) incorporating an anti-racism/anti-colonialism curriculum into promotion packets, (3) recalibrating agenda setting and collaboration, (4) recognizing the importance of mentoring within the promotion packet, and (5) developing effective capacity strengthening initiatives. We summarize the themes at the end of this section by highlighting perspectives from Global South participants in the workshop.

**Making structural changes to the promotion process.** One theme that emerged was structural change within promotion and tenure. Participants identified various areas that were important to change, including change needed in the implicit overall global health climate (e.g. unwritten rules or cultural mindset) and change needed in explicit academic structures, such as department promotion and tenure policies.

Some participants emphasized that focusing on cultural or mindset shifts would be helpful in shifting the promotion processes to value Global North-South partnerships and honor supportive roles in the Global North and leadership from the Global South. During a breakout session, one member stated:

> *"[W]e need to focus on department chairs and what department chairs articulate and mentor to their faculty on what is valued and important, and the chairs represent that...Chairs should be 'socialized' to these set of values."*

Another member in a different breakout session suggested that a *"culture shift [that is] open and accessible"* is necessary to changing the promotion criteria. While department chairs and deans can influence this process, multiple participants believed it would be more powerful if this attitude shift came from a *"larger and more centralized institution"* (such as CUGH). A third participant added "something *strong from CUGH could be that guiding light that we can take back to our institutions and say, 'Here is a global body that has put this forward'.*

Regarding explicit structural changes, participants recommended adjusting department guidelines. Specifically, recommendations centered on how departments defined what "counts" as an indicator for promotion within the promotion packet. Once defined, departments can broaden these indicators to include value for supportive roles of Global North faculty involved with Global South research partnerships. Further, participants stressed there is a strong need to *"educate the [promotion] committees about...new criteria."* This, in turn, pointed back to implicit structural changes that will need to be realigned after the explicit changes are made. During a breakout session, one member stated:

> *"[We need to] recogniz[e] that in addition to changing the promotion criteria, we need to make sure that the promotion committee [that] is reviewing the packet is reoriented towards these efforts and criteria in a proper way. [By] setting one's expectations or criteria, there need to be guidelines and supports for people to act on."*

Other members expanded on this sentiment by recommending that an institution's *"faculty senate could be leveraged"* in codifying promotion guidelines. By clearly defining the current guidelines through these vectors, departments can educate promotion committees on promotion expectations and create a roadmap for the promotion process. Once codified, these guidelines could serve two purposes: as an accountability tool to ensure individuals are following the standards, and to help develop a sense of peer pressure if individuals are not following standards. Finally, these guidelines should be institutionalized and promoted by deans and chairs to set department values. In other words, updated guidelines need to be codified, and the codification must be a collaborative group effort.

Further, members of one breakout group *"acknowledged the hidden labor that...contributes to [faculty] efforts."* Consistently, members stated that promotion and tenure unevenly values publication submissions in relation to other faculty activities. Promotion packets could *"include a statement from the candidate about their contribution to decolonizing global health"* and these packets could be extended to include *"external letters from partners in the Global South."* Incorporating this kind of statement would help *"tell the story behind the grants, publication, and other work"* and explain the roles faculty play in partnerships for inclusion into their promotion packets.

**Incorporating an anti-racism/anti-colonialism curriculum into the promotion packet.** The discussions also focused on ways to incorporate anti-racism pedagogy into faculty teaching promotion packets. Participants called for departments to *"provide guidance for incorporating anti-racism and anti-colonialism towards one's teaching."* Members discussed that teaching anti-racist and anti-colonialist incorporated principles ensures that faculty are making contributions to changing problematic structures for future generations working in global health. By *"incorporat[ing] anti-racism and Diversity, Equity, and Inclusion into promotion criteria,"* leaders in public health can provide *"guidelines and supports for people to act on."*

**Recalibrating agenda setting and collaboration.** Agenda-setting was another theme that arose from the analysis; specifically, where Global North researchers dictate what gets researched in the Global South. As one participant put it, *"We run after the money because that pays our salaries...the funding institutions have to rethink their model of how they fund and how they want to see this happen."* This participant highlighted how much money dictates the agenda. While both Global North and Global South investigators must 'follow the money', this model creates an imbalance. This imbalance is at the expense of Global South partners who have very little say in what drives the projects that fund their salaries, that affect their livelihood, and their communities. Participants also mentioned the imbalance of power Global South researchers have in this model:

> *"My experience [has shown that]...really excellent scientists tend to get caught up in [Global North researchers'] projects... this is a disservice to...them...[and] undermines their career growth."*

What may start as a collaborative effort can quickly become inequitable by focusing on Global North priorities over Global South. Participants also mentioned this phenomenon by stating that Global North researchers were more likely to dictate their research priorities whereas Global South researchers are less inclined to do so:

> *"I have so many examples of North American faculty approaching me and saying, 'I want to work on X. Can you find me a collaborator?' And very few Kenyan collaborators coming up to me and saying, 'I want to work on Y. Can you find me someone in North America that wants to work on that?' That is something that I want to see changed."*

This, in addition to the monetary resources that the Global North brings, often produces a one-sided exchange that systematically disadvantages one group while advancing another.

During the discussion, the co-facilitators presented data which included, "Ideas not yet implemented: credit and recognition." One of these ideas included, *"Valuing first authorship of underrepresented and [Global South] partners."* Having first authorship in journal submissions is highly valued, specifically towards academic research and professional careers in both the Global South and Global North. As such, participants proposed an alternative to move Global

North institutions towards valuing supportive author order placements (second or later) for their investigators on papers from Global North-South collaborative projects.

**Recognizing the importance of mentoring within the promotion packet.** Continuing the theme of empowerment, facilitators noted the importance of *"recognizing mentorship and training of research collaborators, of partners in [Global South] settings."* Many members viewed diversity, equity, and inclusion in the promotion process as a series of parts or components that acted together; where mentoring was an important aspect. Throughout the discussion, participants referred to two versions of proactive mentoring: 1) bilateral mentorship between the Global North and Global South and 2) investigators from the Global South providing mentorship to their local colleagues.

**Developing effective capacity strengthening initiatives.** While it is important to recognize the importance of mentoring, another area that some participants focused on took a macro approach. For example, one member of the workshop clearly expressed, "*... capacity building has to be one of the major components, if we are to ensure a very equitable approach, of research partnerships."* In addition, a Global South respondent who currently works and lives in a Global North country mentioned the importance of capacity building back in their country of origin, *"We are connected to our countries [of origin], and it's...important... to provide these tools [and] funds to go back to our countries and work and develop some of these initiatives."* Not only is capacity strengthening essential, but when it is reinforced by Global South researchers, it is empowering. By bolstering equitable partnerships, furthering the education of the clinical workforce, and empowering trainees, capacity strengthening can be integrated into the entire procedure. These steps can add a powerful dimension to the promotion and tenure process and offer avenues of growth for faculty and their institutions. Much like incorporating anti-racist and anti-colonial perspectives into in global health pedagogy, capacity strengthening efforts impact the future of how global health work will be operationalized. Therefore, it is important that capacity strengthening efforts are clearly recognized within the promotion and tenure process.

While there was much discussion on the various ways to address and dismantle colonization practices from Global North scholars, Global South scholars emphasized a sense of agency needed within this process. During the workshop, one Global South investigator acknowledged that the Global South has nowhere near the resources available compared to the Global North, which led them to feel powerless. Despite working in a Global North institution, one participant who maintains citizenship in the Global South articulated the need for leadership strengthening. They advocated a need to create structures within global health where Global South investigators can take the lead. Global South participants also discussed grassroots methods, where they forge opportunities to collect tools and funding, and in turn, share these resources with others in the Global South through various initiatives and research projects.

## Discussion

Throughout the workshop, participants insisted that structural change needs to occur for equity to exist within global health research. This was brought up alongside needs for changes to the promotion and tenure process within Global North participants' institutions. Further analysis refined this discussion into the categories of explicit structural changes (updating promotion and tenure guidelines within departments) and implicit structural changes (creating an equity-focused mindset within departments). Moreover, within the promotion and tenure discussion, participants argued for developing an anti-racist/anti-colonialist curriculum that faculty can clearly demonstrate in their packets. Participants suggested a shift within their collaboration with Global South partners through their agenda setting. Aligned with many journals' calls for authorship equity,

participants in our workshop also called upon global health researchers to encourage Global South first authorship in papers, while recognizing the challenges achieving this.[28] Finally, departments can begin to institutionalize equity practices within their promotion and tenure process by recognizing mentoring and capacity strengthening partnerships. By integrating these components throughout the promotion and tenure process, faculty will be encouraged to find ways to collaborate equitably with their Global South partners for generations to come. These discussions aligned with the research published about inequities in global health research where hidden systems and Global North agendas are prioritized. (2) Subsequent to these discussions, CUGH put out a position statement that takes into account many of the principles discussed here [29]. Finally, while it was not directly mentioned in the recorded discussion, there needs to be a change within the National Institute of Health (NIH) policy to increase indirect cost recovery for Global South institutions. Currently this is only at 8%, an amount that significantly limits strengthening institutional capacity for research administration. Addressing this change can lead to more funds and resources to help build high quality research administrative capacity in the Global South for Global South investigators. Recently the Fogarty International Center announced that $1.7 million has been earmarked to be distributed for administrative oversight and furthering the goal of developing more diverse global health institutions [30]. Finally, some institutions involved in facilitating this workshop have begun to revise faculty promotion handbooks, and others have begun to develop Global South partner advisory boards to drive their collaborative work. While these are steps in the right direction, much more needs to be done.

We wish to acknowledge that there is some limitation in this study, specifically regarding the sample size. While we had 33 attendees actively participating, many studies have included a larger sample to obtain a richer level of data, but we were limited to participants who chose to attend a pre-conference workshop. Thus, these findings have limited generalizability. Additionally, the sample size had a disproportionate number of Global North investigators to Global South, reflective of CUGH member institution locations. Despite our outreach efforts, there were issues with time zone inconveniences related to time of day that this session was offered. Finally, we noted in the large group discussion that Global North participants were more vocal, and Global South scholars shared more during the smaller breakout sessions. Thus, we must also acknowledge that structural inequities were present within the workshop discussions, as well. Future discussions on this topic would benefit from an awareness of these challenges, and benefit from a plan to challenge these limitations.

## Conclusions

Despite the honorable intentions of global health researchers, several practices within the discipline have been problematic, with unequal power dynamics occurring throughout research processes that are maintained by traditional academic structures and funding constraints. Global North researchers are incentivized to uphold current practices and thus, unintentionally undermine Global South investigators' opportunities and advancement. This paper presents initial discussions of some potential pathways towards structural change in order to change the Global North dominated system. We recommend that institutions begin implementing strategies mentioned here to test their effectiveness.

## Acknowledgements

We wish to acknowledge and thank the participants of the CUGH Virtual Satellite Session Incorporating anti-racism, anti-colonialism, and anti-oppression Efforts into Faculty Promotion for their insight and the CUGH Consortium for their continued support of this important work.

## Author contributions

**Conceptualization:** Deepa Rao, Rae Jean Proeschold-Bell, Bethany Hedt-Gauthier, Jenell Stewart.

**Data curation:** Jade Pearce.

**Formal analysis:** Jade Pearce.

**Investigation:** Jade Pearce.

**Project administration:** Jade Pearce.

**Supervision:** Deepa Rao.

**Validation:** Deepa Rao, Rae Jean Proeschold-Bell, Keshet Ronen, James Pfeiffer, Kearsley A. Stewart, Joao Vissoci, Bryan J. Weiner, L Gayani Tillekeratne, Jenell Stewart.

**Writing – original draft:** Jade Pearce.

**Writing – review & editing:** Jade Pearce, Deepa Rao, Rae Jean Proeschold-Bell, Keshet Ronen, James Pfeiffer, Kearsley A. Stewart, Joao Vissoci, Bryan J. Weiner, L Gayani Tillekeratne, Jenell Stewart.

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
