## [Decision Letter · Decision Letter 0]

30 Oct 2024

PGPH-D-24-01553

Exploring Supportive Roles for North Investigators in Global Health Research

Dear Dr. Pearce,

Thank you for submitting your manuscript to PLOS Global Public Health. After careful consideration, we feel that it has merit but does not fully meet PLOS Global Public Health’s publication criteria as it currently stands. Therefore, we invite you to submit a revised version of the manuscript that addresses the points raised during the review process.

As the Academic editor, I was able to secure only one reviewer for your manuscript. Several reviewers declined the invitation, and while two initially agreed, they ultimately did not provide their feedback. To avoid further delays in the review process, I’ve decided to proceed with sharing the response based on the feedback from this single reviewer.

As mentioned by reviewer 1, I would ask the authors to make the following changes:

Provide a brief overview of the tenure process in institutions in the Global NorthExpand on the conceptual framework in your Discussion section: How do partners from the Global South or pre-workshop participants conceptualize decolonization and anti-racist praxis?Can you tell us more about the following points in the Discussion section : How are you involved in the decolonization movement? And how does your research contribute to decolonization processes (in theory and practice) in the field of Global Health?Could you elaborate on the group dynamics and discussions during the pre-workshop sessions? Were there any diverging perspectives, and what insights do these offer into the ongoing decolonization process?

In my opinion, these additions will greatly enhance the impact of your manuscript.

We look forward to receiving your revised manuscript.

Kind regards,

Marie Meudec, PhD

Academic Editor

Journal Requirements:

 1. In the online submission form, you indicated that "The data in this study is comprised of semi-structured focus groups comprising and is currently unavailable and without restriction at this time due to potentially identifying participant information. Upon request, please contact Jade Pearce at jdpearce@uw.edu for data access.".  All PLOS journals now require all data underlying the findings described in their manuscript to be freely available to other researchers, either 1. In a public repository, 2. Within the manuscript itself, or 3. Uploaded as supplementary information. This policy applies to all data except where public deposition would breach compliance with the protocol approved by your research ethics board. If your data cannot be made publicly available for ethical or legal reasons (e.g., public availability would compromise patient privacy), please explain your reasons by return email and your exemption request will be escalated to the editor for approval. Your exemption request will be handled independently and will not hold up the peer review process, but will need to be resolved should your manuscript be accepted for publication. One of the Editorial team will then be in touch if there are any issues.

Additional Editor Comments (if provided):

Reviewers' comments:

Reviewer's Responses to Questions

**Comments to the Author**

1. Does this manuscript meet PLOS Global Public Health’s publication criteria ? Is the manuscript technically sound, and do the data support the conclusions? The manuscript must describe methodologically and ethically rigorous research with conclusions that are appropriately drawn based on the data presented.

Reviewer #1: Yes

2. Has the statistical analysis been performed appropriately and rigorously?

Reviewer #1: N/A

3. Have the authors made all data underlying the findings in their manuscript fully available (please refer to the Data Availability Statement at the start of the manuscript PDF file)?

Reviewer #1: Yes

4. Is the manuscript presented in an intelligible fashion and written in standard English?

Reviewer #1: Yes

5. Review Comments to the Author

Reviewer #1: Thank you for addressing this important issue regarding decolonizing global health partnerships. Overall, the paper reads well and provides a comprehensive literature review, clearly described methods and findings. The authors engaged with relevant literature on this topic. I have two main suggestions. The first is that the section on promotion practices in the findings section could benefit from a brief description of what the tenure process in global north institutions entails. This may be useful to readers who are unfamiliar with the tenure process in the U.S., for instance and would provide important context for the reader. My second question pertains to the Discussion section. The authors mostly summarize their main findings and do not offer a critical analysis of their findings. I think the paper could benefit from an intellectual reflection on how global south partners or the pre-workshop participants understand decolonization and anti-racist praxis. I think that some theorizing of the concept of decolonization would strengthen the conceptual contributions of the paper. As it currently stands, the paper is largely descriptive and summarizes the key themes that emerged from the pre-workshop. Furthermore, we are not really provided any insights into the tone of the pre-workshop discussions: was the mood one of general consensus or was there some debate around some of these issues? Getting a sense of some of the areas of divergence could be illuminating. There have been some critiques of the "decolonization moment" and it would be great for the authors to engage with some of these in their Discussion.

6. PLOS authors have the option to publish the peer review history of their article (what does this mean? ). If published, this will include your full peer review and any attached files.

**Do you want your identity to be public for this peer review?** For information about this choice, including consent withdrawal, please see our Privacy Policy .

Reviewer #1: No

---

## [Decision Letter · Decision Letter 1]

11 Feb 2025

Exploring Supportive Roles for Global North Investigators in Global Health Research

PGPH-D-24-01553R1

Dear Dr. Rao,

We are pleased to inform you that your manuscript 'Exploring Supportive Roles for Global North Investigators in Global Health Research' has been provisionally accepted for publication in PLOS Global Public Health.

Best regards,

Zahra Zeinali, MD MPH DrGH (c)

Academic Editor

Reviewer Comments (if any, and for reference):

Reviewer's Responses to Questions

**Comments to the Author**

1. If the authors have adequately addressed your comments raised in a previous round of review and you feel that this manuscript is now acceptable for publication, you may indicate that here to bypass the “Comments to the Author” section, enter your conflict of interest statement in the “Confidential to Editor” section, and submit your "Accept" recommendation.

Reviewer #1: All comments have been addressed

2. Does this manuscript meet PLOS Global Public Health’s publication criteria ? Is the manuscript technically sound, and do the data support the conclusions? The manuscript must describe methodologically and ethically rigorous research with conclusions that are appropriately drawn based on the data presented.

Reviewer #1: Yes

3. Has the statistical analysis been performed appropriately and rigorously?

Reviewer #1: N/A

4. Have the authors made all data underlying the findings in their manuscript fully available (please refer to the Data Availability Statement at the start of the manuscript PDF file)?

Reviewer #1: Yes

5. Is the manuscript presented in an intelligible fashion and written in standard English?

Reviewer #1: Yes

6. Review Comments to the Author

Reviewer #1: The authors have responded to my previous concerns satisfactorily. I recommend the paper for publication.

7. PLOS authors have the option to publish the peer review history of their article (what does this mean? ). If published, this will include your full peer review and any attached files.

**Do you want your identity to be public for this peer review?** For information about this choice, including consent withdrawal, please see our Privacy Policy .

Reviewer #1: No
